# Preventive Healthcare and Disability: Challenges and Opportunities

**DOI:** 10.3390/healthcare13172099

**Published:** 2025-08-23

**Authors:** Giovanni Emanuele Ricciardi, Rita Cuciniello, Veronica Raimondi, Francesco Vaia, Carlo Signorelli, Cristina Renzi

**Affiliations:** 1School of Medicine, Vita-Salute San Raffaele University, 20132 Milan, Italy; ricciardi.giovanni@hsr.it (G.E.R.); cuciniello.rita@hsr.it (R.C.); raimondi.veronica@hsr.it (V.R.); signorelli.carlo@hsr.it (C.S.); 2San Raffaele Hospital, 20132 Milan, Italy; 3PhD National Programme in One Health Approaches to Infectious Diseases and Life Science Research, Department of Public Health, Experimental and Forensic Medicine, University of Pavia, 27100 Pavia, Italy; 4National Authority for the Rights of Persons with Disabilities, 00175 Rome, Italy; f.vaia@garantedisabilita.it; 5National Institute for Infectious Diseases “Lazzaro Spallanzani” IRCCS, 00149 Rome, Italy; 6Research Department of Behavioural Science and Health, University College London, London WC1E 6BT, UK

**Keywords:** disability, preventive health services, cancer screening, vaccination, infection prevention, health equity, barriers to access, inclusion, universal health coverage, accessibility

## Abstract

Despite global commitments to universal health coverage, persons with disabilities (PwD) continue to face significant barriers in accessing appropriate healthcare, including diagnostics, treatments and preventive healthcare, with lower participation in cancer screening and vaccination programs. These disparities are driven by diverse, intersecting obstacles (structural, financial, communicative, and social) that vary by disability type and context. Inclusive approaches, co-designed with PwD and supported by standardized assessment tools, are urgently needed to address persistent inequities in healthcare access and outcomes.

## 1. Introduction

Sustainable Development Goal (SDG) 3 of the United Nations 2030 Agenda aims to “ensure healthy lives and promote well-being for all at all ages.” Among its specific targets, Target 3.8 calls for achieving universal health coverage (UHC), meaning that all individuals and communities should have access to the full spectrum of quality health services they need, including preventive, promotive, curative, rehabilitative, and palliative care, without suffering financial hardship. Article 25 of the Convention on the Rights of Persons with Disabilities (CRPD) reinforces the right of persons with disabilities (PwD) to attain the highest standard of health without discrimination, ensuring equal access to general medical care and disability-specific services, including preventive services [1].

The CRPD defines disability as a long-term condition characterized by physical, mental, intellectual, or sensory limitations resulting from one or more impairments, leading to a reduced ability to engage with the social environment and decreased autonomy in daily activities. PwD represent a significant and growing proportion of the world’s population. According to the World Health Organization (WHO), an estimated 1.3 billion individuals globally live with disabilities, representing 16% of the worldwide population [2].

Despite this global commitment, PwD continue to encounter substantial barriers to healthcare access, particularly preventive services and essential medical care [3]. These barriers contribute to persistent health inequities and highlight the need for inclusive strategies to truly “leave no one behind” in the pursuit of UHC.

The evidence presented in this commentary draws from a targeted literature review conducted through searches in PubMed, Scopus and Web of Science databases using terms related to disability, preventive healthcare, cancer screening, and vaccination. Priority was given to systematic reviews, meta-analyses, and recent primary studies from peer-reviewed journals, with attention to evidence from diverse geographic and economic contexts.

## 2. Participation in Cancer Screening Programs

Cancer screening is a cornerstone of secondary prevention and early detection and is recognized as a fundamental approach for reducing cancer morbidity and mortality. Nevertheless, evidence shows that PwD are less likely to participate in cancer screening programs compared to the general population [4]. Recent research on routes to colon cancer diagnosis has suggested that patients with chronic conditions, such as dementia [5] or mental health conditions [6], are less likely to be diagnosed with cancer through screening and more likely to be diagnosed as an emergency, with worse outcomes. In the case of colorectal cancer, this gap is further exacerbated when patients experience red-flag symptoms, which should prompt diagnostic investigation, which are sometimes incorrectly attributed to pre-existing conditions or treatments [7,8]. Such misattribution can lead to delays in appropriate diagnostic procedures. However, the evidence is limited, and findings vary depending on the type of disability, suggesting that the relationship between disability status and routes to cancer diagnosis is complex and influenced by a variety of factors.

A recent systematic review and meta-analysis showed that women with disabilities have a 22% reduced odds of undergoing mammographic screening (OR 0.78; 95%CI: 0.72–0.84) and a 37% reduced odds for cervical screening (OR 0.63; 95%CI: 0.45–0.88) compared to women without disabilities [9]. The studies primarily focused on psychosocial disability (47% of the sample), followed by general or combined disability (31.3%). Only a few studies separately examined specific intellectual, physical, visual or hearing disabilities. Subgroup analyses reveal even greater disparities for certain types of disability: women with visual impairments have a 37% reduction in the likelihood of mammographic screening (OR: 0.63; 95%CI: 0.51–0.77), and those with psychosocial disabilities show a 31% decrease (OR: 0.69; 95%CI: 0.60–0.80).

Similarly, our meta-analysis on colorectal cancer screening found that PwD are 20% less likely to participate than those without disabilities (OR 0.80; 95%CI: 0.73–0.87). This lower participation was particularly evident for fecal occult blood tests (FOBT) or fecal immunochemical tests (FIT), with PwD showing a lower likelihood of completing these tests (OR: 0.72, 95%CI: 0.65–0.81) [10]. Subgroup analyses for colorectal cancer screening were also conducted based on specific types of disability. Individuals with functional disabilities were significantly less likely to be screened with FOBT/FIT, compared to those without disabilities (OR: 0.59, 95%CI: 0.47–0.73). Similarly, people with visual impairments were less likely to undergo any type of colorectal cancer screening (OR: 0.74, 95%CI: 0.61–0.89). Those with intellectual disabilities had markedly lower rates of colorectal cancer screening overall (OR: 0.65, 95%CI: 0.53–0.79), with an even lower likelihood for FOBT/FIT tests specifically (OR: 0.58, 95%CI: 0.49–0.69). People with psychosocial disabilities also demonstrated reduced participation in any colorectal cancer screening (OR: 0.82, 95%CI: 0.69–0.97).

The reduced participation rates have significant clinical implications. This suggests that cancers in this population are likely to be diagnosed at later stages, which can result in poorer survival rates, more aggressive treatments, and fewer opportunities for less invasive interventions. Ultimately, these disparities contribute to preventable excess mortality and highlight the urgent need for accessible and disability-inclusive screening pathways.

Both reviews highlighted a fundamental limitation: all studies originated from high-income countries, particularly the United States. This severely limits the generalizability of findings to low- and middle-income countries, where the majority of the global disabled population resides. Additionally, both reviews acknowledge the high degree of heterogeneity among the included studies, with significant variations in disability definitions and assessment methodologies. This methodological inconsistency fundamentally compromises the ability to synthesize findings and draw robust conclusions across studies. The lack of standardized approaches not only affects the consistency and comparability of results but also limits the development of evidence-based guidelines for screening practices in disabled populations.

## 3. Participation in Immunization Programs

Participation in immunization programs is essential for preventing infectious diseases and reducing serious complications, as well as for preventing some infection-associated cancers. However, data show that adherence to vaccination campaigns remains sub-optimal in children and adolescents with disabilities. Studies have shown that individuals with a physical, neurological, or intellectual disability have a higher risk of hospitalizations, severe complications, and death due to several preventable infections, including chicken pox and flu [11,12]. From a public health perspective, these gaps in vaccine coverage among PwD contribute to a disproportionate burden of preventable disease and associated healthcare costs. However, quantitative data that directly links low vaccination coverage to specific disease outcomes (such as infection, hospitalization, or mortality rates attributable to vaccine-preventable diseases) for the PwD are deficient or difficult to aggregate.

A 2019 review on vaccinations highlighted that PwD have lower immunization uptake rates across various vaccines compared to the general population [13]. There was considerable variation in immunization uptake depending on the type of disability, setting, and vaccine, but 78% of the studies included in the review reported reduced vaccination rates among PwD. Lower vaccination rates have been found for several conditions, including Inborn Errors of Metabolism (IEM), Autism Spectrum Disorders (ASD), intellectual disability, cerebral palsy and spina bifida. Findings are further complicated by substantial variations in disability definitions, cultural contexts, vaccination types, and age group classifications across studies. Notably, studies grouping heterogeneous disability categories, such as Youth with Special Health Care Needs (YSHCN), failed to detect significant differences, suggesting that important variations among specific disability subgroups may be masked by overly broad categorizations [14]. This methodological approach may obscure critical insights about vaccination needs and responses in distinct disability populations. The review also identified a significant gap in qualitative studies that explore the experiences and perspectives of individuals with disabilities concerning vaccination [13]. Additionally, a critical limitation is that most studies originate from high-income countries, with insufficient representation from low- and middle-income countries where most disabled individuals globally live. This geographic bias significantly constrains the applicability of findings across diverse economic settings.

Emerging evidence from low- and middle-income countries reveals particularly concerning disparities in vaccination coverage among PwD. A recent study in Fiji found that only 55% of children with disabilities were completely vaccinated against basic antigens, significantly below the national average, with barriers including inadequate healthcare infrastructure, stigma, and lack of disability-specific knowledge among healthcare providers [15]. Similarly, research in Vietnam demonstrated that children with cerebral palsy had vaccination rates of 82.7% compared to 96.4% in the general population, with incomplete vaccination associated with factors such as home births, poor housing conditions, and low maternal education [16].

The COVID-19 pandemic has emphasized the pre-existing challenges faced by PwD. Many individuals in this group have experienced delays and outright exclusion from crucial priority vaccination categories, despite increased concern about COVID-19 and a greater intention to vaccinate, as revealed by some quantitative and qualitative studies, some of which were carried out in low- and middle-income countries [17,18,19].

## 4. Participation in Other Preventive Services

Recent research indicates that PwD also face substantial disparities in accessing other preventive services beyond cancer screening and immunization, including dental care, mental health screening, and general physical examinations.

A systematic review published in 2024 emphasizes that PwD highly value oral health and dental care, yet they face significant challenges in accessing dental services [20]. The review found that dental care is mainly sought during emergencies. Notably, nearly half (47%) of children with Down syndrome and more than a third (37%) of those with physical disabilities had their first dental visit at the age of 6 or older. Furthermore, only 69.2% of dentists provide care for patients with disabilities, and among those, 73.5% treat fewer than 10 patients with physical disabilities each year. Additionally, 54% of dentists do not treat individuals with cognitive impairments who may struggle to cooperate during treatment. Communication barriers also exist, as 56.2% of dentists find it difficult to communicate with deaf patients, and 97.8% expressed a desire for interpreters. However, although this systematic review included low- and middle-income countries, it was limited by the high degree of heterogeneity among the included studies, with significant variations in disability definitions and assessment methodologies compromising the ability to synthesize results across studies.

Access to mental health screening presents another critical gap in preventive care for PwD. A comprehensive scoping review found that PwD are significantly more vulnerable to developing depression than the general population, with prevalence rates ranging from 8.06% to 100% depending on the type of disability and measurement tools used [21]. The review identified that the Patient Health Questionnaire (PHQ-9) was the most widely used depression screening tool (29.26% of studies), though substantial variation in assessment methods limited comparability across studies. Furthermore, access to mental health services for PwD in general indicates that this population has more unmet needs. Notably, this study included studies from 22 countries, with significant representation from low and middle-income countries, providing valuable insights into depression among PwD across diverse economic contexts. However, this scoping review was limited by the predominantly cross-sectional nature of the included studies (78.04%), the lack of evaluation of the quality of randomized controlled trials (RCTs), and the inability to analyze depression status over time due to limited longitudinal data.

Access to general physical examinations shows similarly concerning disparities. A systematic review of 2018 examining healthcare access in low- and middle-income countries found mixed results regarding preventive care coverage for PwD [22]. While utilization of healthcare services was generally higher among PwD (likely due to greater healthcare needs), coverage outcomes varied significantly across different preventive services. The majority of studies (59%) found no difference in general healthcare coverage between people with and without disabilities, though this masked significant variations in specific services and disability types. The review was limited by the wide variation in disability types and access measures across studies, making comparisons difficult, and nearly half of the studies (46%) were judged to have high or moderate risk of bias, which may have influenced findings and limited the reliability of conclusions about preventive care access patterns.

## 5. Barriers and Determinants of Access

Access to preventive healthcare services for PwD is affected by a complex network of interconnected barriers. These barriers, which can be categorized as structural, physical, financial, transportation-related, provider-related, communicative, and socio-cultural, often overlap in real life, leading to significant disparities in health outcomes (Figure 1). Their cumulative impact may vary depending on the type and severity of disability.

Structural and physical barriers remain among the most pervasive obstacles. Many healthcare facilities are not fully accessible, with architectural features and a lack of adapted medical equipment making it difficult for individuals with mobility impairments to undergo examinations or access diagnostic procedures [23]. For example, research in Kenya found that persons with physical disabilities faced significant challenges accessing COVID-19 vaccination sites due to inaccessible facilities, lack of ramps, narrow doorways, and inadequate accessible parking [18]. Transportation challenges further exacerbate these difficulties, especially for people living in rural or underserved areas with limited or unreliable public transport [24].

Financial barriers are also significant. The direct costs of healthcare, insufficient insurance coverage, and additional expenses related to disability, such as assistive devices or personal care support, can deter PwD from seeking preventive care or adhering to recommended screening protocols [25]. Qualitative studies have highlighted that financial constraints and apprehensions about procedures such as bowel preparation for colonoscopy and sigmoidoscopy can strongly dissuade participation in preventive programs [26]. Research on people with visual impairment has specifically identified increased difficulties in accessing healthcare due to cost and lack of insurance coverage [27].

Provider-related factors are crucial, as many healthcare professionals lack adequate training in disability-specific care. This deficit can lead to inadequate communication, discomfort, or even stigma and discriminatory attitudes, resulting in missed preventive opportunities and negative patient healthcare experiences [28]. An exploratory qualitative study of medical specialists revealed a fundamental uncertainty about professional responsibility, with many questioning whether caring for people with intellectual disabilities was their job [29]. The research found that stigma and limiting representations about the abilities of people with intellectual disabilities often lead professionals to generalize emergencies or legal exceptions to all patients in this population, systematically failing to obtain informed consent from the person themselves.

Communication barriers are particularly pronounced for individuals with sensory or intellectual disabilities, as the absence of accessible information, such as sign language interpreters or easy-to-read materials, can impede understanding and participation in preventive initiatives. Research on cancer screening among deaf, deafblind, and hard-of-hearing adults demonstrated the critical need for specialized community health navigators to overcome communication barriers [30]. Stigma and limiting representations about the abilities of people with intellectual disabilities lead professionals to generalize emergencies or legal exceptions to all, failing to obtain informed consent from the person [29]. This has profound legal and ethical implications and requires specialists to be informed about the legal status of persons with intellectual disabilities and their duty to obtain informed consent from the person.

Socio-cultural determinants further compound access issues and contribute to cumulative disadvantage [25]. These determinants include low health literacy, internalized stigma, lack of social support, and reduced self-advocacy skills. For PwD, the reluctance to seek assistance may be due to fear of not receiving care appropriate to their needs. They may feel vulnerable to discriminatory actions stemming from stereotypes [23]. Qualitative research indicates that PwD often face inadequate knowledge of preventive recommendations, language obstacles, logistical challenges, and cultural beliefs that affect their adherence to screening protocols [31]. For example, research in Fiji documented how stigma and lack of disability-specific knowledge among healthcare providers created significant barriers to vaccination access for children with disabilities, contributing to vaccination rates of approximately half compared to national averages [15].

Significantly, the nature and intensity of these barriers can vary according to the type of disability. For instance, current communication strategies often fail to meet the needs of individuals with intellectual disabilities [32]. Studies focusing on psychosocial disabilities have documented significant disparities in access to secondary cancer prevention, which are associated with a marked reduction in life expectancy compared to the general population [33]. Similarly, research on people with visual impairment has identified increased difficulties in accessing healthcare due to cost, lack of insurance coverage, transportation problems, and even service refusal by providers [27].

It is important to recognize that this categorization, while useful for analysis and intervention planning, does not reflect the seamless way these barriers intersect in daily life. A single healthcare encounter may simultaneously involve multiple barrier types: financial constraints limiting transport options, inaccessible facilities requiring provider accommodation, and communication challenges compounded by cultural misunderstandings. These determinants frequently overlap, resulting in a cumulative effect that increases the risk of preventable diseases and poorer health outcomes among PwD. Addressing these multifaceted barriers requires a comprehensive and tailored approach, informed by quantitative and qualitative research.

## 6. Towards Inclusive Strategies, Policy and Future Directions

To overcome the barriers described above and promote the genuine inclusion of PwD in preventive services, integrated strategies must be adopted at the health system, public policy, and community levels. An essential first step is to ensure the physical and organizational accessibility of health facilities by adapting spaces, using appropriate equipment, and guaranteeing accessible pathways for all types of disabilities [2]. All providers within the territory must be involved in this process.

In Italy, several strategies have been adopted to increase vaccination coverage, each addressing specific logistical, social, and structural challenges [34,35]. These include a mixed delivery model involving health services offered by public and private providers [36]. At the same time, ongoing training of health workers on disability and accessible communication is crucial to improving the quality of care and reducing discriminatory attitudes. Practical examples are home-based testing options or drop-off zones to reduce walking [18], communication adaptations for sensory disabilities and training programs focusing on the “See-Hear-Feel-Speak” protocol and universal design principles [37]. Evidence-based adaptations have also enhanced cancer screening for PwD. In Australia, a consensus model using “Easy Read” materials, peer mentors, and service navigators improved mammography uptake among women with intellectual disabilities [38].

From a policy perspective, integrating the disability viewpoint into health planning is essential. This integration requires actively involving PwD and their representative organizations in the programming processes for preventive services. In this context, the implementation of independent monitoring mechanisms, as Article 33 of the CRPD requires, plays a crucial role [1]. Many countries have responded to this obligation by establishing national frameworks or authorities tasked with safeguarding the rights of PwD. For example, Italy recently established a National Authority for the Rights of Persons with Disabilities, specifically mandated to promote, protect, and monitor the rights and freedoms of PwD, including their active participation in policy development. Similarly, other countries have set up independent monitoring bodies, often within national human rights institutions or dedicated committees, to ensure that the voices of PwD and their representative organizations are meaningfully included in health planning and service delivery.

For instance, Germany designated the German Institute for Human Rights as its independent monitoring mechanism, France established the Défenseur des droits operating through a network including 100 specialized disability delegates, and the United Kingdom adopted a multi-institutional approach through the UK Independent Mechanism comprising regional human rights bodies. India established a Chief Commissioner for Persons with Disabilities under its 2016 law to oversee implementation and coordinate agencies. In Africa, Nigeria’s National Human Rights Commission serves as the CRPD monitoring body, Kenya’s National Commission on Human Rights leads a CRPD National Action Plan, and South Africa designated its Human Rights Commission as an independent CRPD monitor with a dedicated Disability Toolkit. However, many low- and middle-income countries struggle with limited funding, weak institutional capacity, poor inter-agency coordination, and ensuring meaningful participation by disability organizations.

Enhancing scientific research is crucial, particularly in low- and middle-income countries, to gain a comprehensive understanding of the unique challenges faced by PwD and to identify the most effective strategies for inclusion in preventive health services. Addressing existing knowledge gaps requires not only a greater quantity of studies but also improvements in research quality and representativeness. The active involvement of PwD at every stage of the research process, including study design, implementation, and dissemination of results, is essential to ensure that interventions are relevant and grounded in lived experience. Another important consideration is how disability is defined within studies. This aspect can directly shape both practice and policy: the choice of assessment tool determines which individuals are identified, which subgroups are represented, and ultimately, who benefits from preventive services. If inappropriate or inconsistent tools are used, significant populations may be overlooked, data comparability is hindered, and resources may be misallocated. Conversely, standardized and context-sensitive assessment instruments enable accurate monitoring, fair resource allocation, and the development of interventions that truly address the needs of all PwD. While many researchers rely on clinical diagnoses to identify issues such as vision or hearing impairments and mental health conditions, some studies depend on self-reported disabilities without clarifying the specific type. Typically, disability is measured by asking individuals to report on their functional abilities, such as seeing, hearing, or moving, in line with the International Classification of Functioning, Disability and Health (ICF). Several validated instruments are available for assessing disability, including the Washington Group Short Set of Functioning (WG-SS), the Washington Group Short Set of Functioning—Enhanced (WG-SS Enhanced), the Washington Group Extended Set of Functioning (WG-ES), the Model Disability Survey (MDS), the Equality Act Disability Definition (EADD), and the Global Activity Limitation Instrument (GALI). These instruments are widely implemented across diverse contexts: the WG-SS is used by countries such as South Africa [39], Cameroon, Guatemala, India, Maldives, Nepal, Turkey and Vanuatu [40], while the MDS serves as the WHO and World Bank’s recommended standard for comprehensive disability data collection in countries such as Brazil [41], Chile, and Sri Lanka [42]; the EADD is primarily employed within the UK’s healthcare and public sectors [43], and the GALI functions as the official European Union indicator used by Eurostat and 22 EU member states for monitoring health policies and calculating Healthy Life Years [44]. Each assessment tool has its strengths and limitations (Table 1). However, the choice of measurement instrument can significantly impact disability prevalence estimates, with research demonstrating that selecting one tool over another can shift prevalence rates by double-digit percentages, complicating cross-study comparisons and potentially affecting resource allocation and policy decisions [45,46,47]. Furthermore, the standardization inherent in these instruments may flatten the heterogeneity that makes each disabled person’s pathway to exclusion distinct, potentially obscuring the unique individual experiences and diverse barriers that characterize disability.

## 7. Conclusions

In conclusion, realizing the principles of the Sustainable Development Goal 2030 Agenda demands more than high-level commitments; it calls for coordinated, integrated action across sectors, grounded in practical steps. This means actively engaging communities, especially those often marginalized, in designing and implementing health and social programs. Expanding access to essential services through outreach initiatives, mobile health units, and digital solutions can help bridge existing gaps. At the same time, targeted education and awareness campaigns are needed to address socio-cultural, emotional, and legal barriers, empowering individuals to understand and exercise their rights. However, partnerships between governments, civil society, and the private sector must acknowledge inherent power imbalances and competing interests. Effective accountability requires clear standards, regular monitoring by independent bodies, including organizations of PwD, and transparent reporting of both successes and failures. When private providers fail to meet accessibility standards or when public authorities reduce services due to budget constraints, there must be mechanisms for redress and enforcement. Progress should be monitored through transparent data collection and public reporting, ensuring accountability and timely adjustments. Strong partnerships can combine resources and expertise, but only when backed by concrete commitments, measurable targets, and consequences for non-compliance. To translate our analysis into practice, we recommend:Ensuring physical and organizational accessibility of all preventive health facilities.Incorporating disability-specific training and accessible communication protocols for all health workers.Actively involving PwD and their representative organizations in service design and policy planning.Standardizing and sensitizing disability assessment tools to inform equitable resource allocation and monitoring.

These concrete actions bring us closer to a future where no one is left behind and everyone can lead a healthy, fulfilling life.

## Figures and Tables

**Figure 1 healthcare-13-02099-f001:**
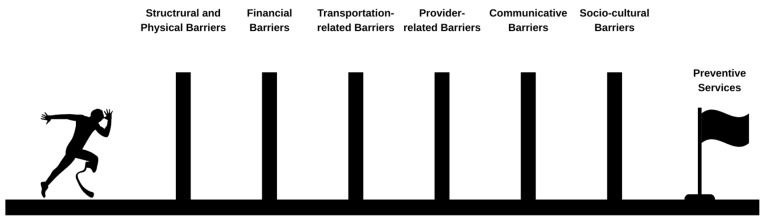
Barriers to access preventive services for PwD.

**Table 1 healthcare-13-02099-t001:** Key strengths and limitations of common disability assessment tools used in population surveys and health-service planning.

Approach	Strengths	Limitations
Clinical Assessment	Provides relevant data for health and rehabilitation service planning	Need for clinical equipment or personnel may increase the resource burden
WG-SS	Recommended for SDG Data disaggregation Simple and quick Internationally comparable	Misses psychosocial functioning Does not measure participation restrictions
WG-SS Enhanced	Includes psychosocial functioning and upper body domains Inherits WG-SS advantages	Does not measure participation restrictions
WG-ES	Assesses the full spectrum of functioning (including participation) Allows detailed analysis of environmental barriers and facilitators	Requires trained interviewers Longer administration time than WG-SS
MDS	Comprehensive survey methodology Provides detailed information	Requires complex statistical analysis Time-consuming and resource-intensive compared to alternatives
EADD/GALI	Concise Includes participation restrictions	Does not give details on functional domains Does not specify whether the limitation is with or without assistive devices or medication

WG-SS: Washington Group Short Set of Functioning; WG-SS Enhanced: Washington Group Short Set of Functioning—Enhanced; WG-ES: Washington Group Extended Set of Functioning; MDS: Model Disability Survey; EADD: Equality Act Disability Definition; GALI: Global Activity Limitation Instrument.

## Data Availability

No new data were created or analyzed in this study. Data sharing is not applicable to this article.

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
