# Peer review of "Preventive Healthcare and Disability: Challenges and Opportunities"

_healthcare, 2025, doi:10.3390/healthcare13172099_

Round 1

Reviewer 1 Report

Comments and Suggestions for Authors

In this manuscript, the authors discuss preventive healthcare for people with disabilities (PwD), outlining the barriers they face in accessing such services and offering policy-level recommendations to improve healthcare inclusion. While the manuscript is informative and relevant, several issues and suggestions are outlined below for further consideration:

  1. The manuscript presents robust data from meta-analyses, with clearly reported odds ratios (ORs) and confidence intervals (CIs), and references that are relevant and up to date. However, as most cited studies originate from high-income countries, the authors are encouraged to incorporate findings from a broader range of sources to enhance the manuscript’s global relevance.
  2. The manuscript focuses on preventive healthcare and disability by highlighting low participation in cancer screening and vaccination programs. The inclusion of other preventive services—such as dental care, mental health screenings, and general physical examinations—would strengthen the comprehensiveness of the discussion, if relevant studies are available.
  3. The authors reference numerous studies to describe participation of people with disabilities (PwD) in cancer screening and vaccination programs. Given the heterogeneity of these studies, presenting them in summary tables—organized by preventive service—could improve comparability. Such tables might include information on population size, disability type, participant demographics (e.g., age and gender), and study location, etc.
  4. The authors identify a wide range of barriers to accessing preventive services for people with disabilities and recommend integrated strategies at the health system and community levels to address them. To strengthen the practical value of the manuscript, it would be helpful to provide more detailed recommendations tailored to specific types of disabilities, for example, implementing sensory-friendly screening environments or offering home-based testing options.

Reviewer 2 Report

Comments and Suggestions for Authors

This commentary addresses the critical intersection of preventive healthcare and disability, examining barriers to cancer screening and vaccination programs for people with disabilities (PwD). The authors synthesize evidence from systematic reviews and meta-analyses to highlight significant disparities in healthcare access, particularly for cancer screening (22% reduced odds for mammography, 37% for cervical screening) and vaccination programs. While the topic is highly relevant to global health equity goals and the manuscript provides valuable insights into a neglected population, several areas require strengthening to enhance the scientific contribution.

Strengths

  1. The manuscript effectively covers multiple aspects of preventive healthcare
  2. Good use of systematic review and meta-analysis evidence
  3. Inclusion of policy recommendations and assessment tools
  4. Generally well-structured and accessible to diverse audiences

Major Concerns

  1. A fundamental limitation that requires more explicit acknowledgment is the predominant focus on high-income countries. As acknowledged in lines 86-92 and 106-108, virtually all studies originated from high-income settings, particularly the United States. This severely limits the generalizability of findings to low- and middle-income countries where the majority of the global disabled population resides. It is recommended that this limitation, along with the need for further research in diverse economic settings, be stated more explicitly.
  2. As acknowledged in lines 86-92 and 106-108, the high heterogeneity among studies requires deeper interpretation and specific recommendations for future research standardization

Specific Comments

Lines 47-92: The cancer screening section effectively synthesizes quantitative evidence but would benefit from more discussion of the clinical implications of these disparities. What does a 22% reduction in mammography screening mean for cancer outcomes in this population?

Lines 93-118: The vaccination section effectively identifies the specific vulnerabilities of people with disabilities; however, it could be strengthened by a deeper analysis of public health implications and the inclusion of more quantitative data on disease burden within this population.

Lines 174-219: The solutions section is well-conceived but lacks specificity. Consider providing more concrete examples of successful interventions or policy approaches from different countries.

Lines 202-218: The discussion of disability assessment tools is valuable but seems somewhat disconnected from the main narrative. Consider integrating this more seamlessly or relocating to a methods consideration section.

The manuscript addresses an important and under-researched topic with significant public health implications. The core contribution is solid, and the synthesis of existing evidence is valuable. However, the limitations regarding geographic scope and methodological heterogeneity require more explicit acknowledgment and interpretation. With appropriate revisions, particularly addressing the generalizability concerns and providing more specific guidance for future research, this commentary could make a meaningful contribution to the disability and public health literature.

The work is technically sound and addresses a critical gap in healthcare equity research, but the current presentation underestimates the implications of its methodological limitations for global health policy and practice.

Reviewer 3 Report

Comments and Suggestions for Authors

This study is a commentary article based on current literature that summarizes the multidimensional barriers experienced by individuals with disabilities in accessing preventive healthcare services.

Minor corrections are acceptable.
1) The article's search strategies (which databases? WoS, Scopus, Pubmed, etc., keywords, and inclusion/exclusion criteria) are not explained. This should be explained.
2) Figure 1 is presented too simply. Obstacles could be created in the form of stairs, and percentages of the total obstacles could be displayed for visual richness.
3) Minor corrections are recommended for methodological details, graphic/table presentation, and limitations.

Reviewer 4 Report

Comments and Suggestions for Authors

This commentary sets out to spotlight the stubborn gap between global pledges of universal health coverage and the day-to-day reality experienced by people with disabilities when they try to access cancer screening or routine immunisation.  In doing so, it performs a valuable service: it gathers, in one place, a sobering array of statistics showing that participation rates for breast, cervical, and colorectal screening are 20–40 % lower among disabled populations, and that vaccine uptake follows a similar pattern.  Yet the same pages also reveal, often unwittingly, the limits of the evidence they marshal and the contradictions built into their prescription.

The authors rely heavily on two recent meta-analyses, one on cancer screening and one on colorectal FIT/FOBT participation, which are welcome additions to the literature. Both, however, are overwhelmingly drawn from the United States and a handful of other high-income settings; the commentary concedes this limitation, but then proceeds as if the patterns will hold once resources are scarcer, stigma is sharper, and health systems are thinner. That leap of faith is nowhere tested.  Likewise, the heterogeneity they lament, different disability definitions, different age bands, and different outcome measures, does not dissolve simply because it is noticed; it undercuts any confident claim of a universal 20% deficit.

A second tension runs through the discussion of barriers. The commentary sketches six tidy boxes: structural, financial, transport, provider, communicative, and socio-cultural, but the lived experience described in the qualitative snippets refuses to stay inside them. Transport hardship bleeds into financial hardship; provider unfamiliarity merges with subtle stigma; a visually impaired woman’s missed mammogram is simultaneously an equipment failure, a scheduling maze, and a reflection of low self-advocacy. The neat typology is rhetorically useful, yet it risks re-enacting the very fragmentation it seeks to repair.

The policy section, meanwhile, oscillates between bold ambition and modest incrementalism. On one page, we are told that “integrated strategies must be adopted at the health-system, policy and community levels”; on the next, we are offered Italy’s mixed public–private vaccine delivery model and a new national disability authority as exemplars. The reader is left to wonder how a country-specific administrative tweak scales to the universalism demanded by the SDGs, especially when the commentary elsewhere insists that every setting is unique and that cultural context is decisive.

Most striking is the silence on measurement politics. The commentary lists six validated disability instruments, ranging from the Washington Group Short Set to the Model Disability Survey, and compares their strengths and weaknesses politely.  But it never confronts the awkward truth that choosing one over the other can shift prevalence rates by double-digit percentages, which in turn can dilute or inflate the very disparities the piece decries. Nor does it grapple with the possibility that standardisation itself may flatten the heterogeneity that makes each disabled person’s exclusionary pathway distinct.

Finally, the commentary ends with a flourish of partnership rhetoric: governments, civil society, and the private sector must “combine resources and expertise.”  Laudable, yet the article offers no mechanism for holding these actors accountable when, for example, a private diagnostic chain refuses to install height-adjustable examination tables, or when a cash-strapped ministry quietly narrows the screening age band to save money. The absence of engagement with power asymmetries renders the call for inclusion more aspirational than actionable.

Though the commentary succeeds as a concise map of the field’s current standing, it also exposes the fault lines beneath the map. Until those fault lines are acknowledged and until the evidence base is broadened beyond its North Atlantic comfort zone, the promise that “no one will be left behind” will remain an elegant slogan rather than a measurable outcome.

Reviewer 5 Report

Comments and Suggestions for Authors

The introduction needs to have its references changed so that they are clearer. For example, reference [1] is used twice in the same paragraph with almost the same information. This might be made easier by combining the two references. There is some repetition in the chapter about taking part in cancer screening programs. For example, the information about women with psychological or visual impairments in mammographic screening (p. 2, lines 68–72) is the same as what was said in the previous paragraph. I think it would be best to put these results into a short summary.
The part about vaccination programs mostly talks about studies done in high-income countries, especially the US and the UK. It would be a good idea to include examples or data from nations with low or intermediate incomes. Also, there are no particular suggestions for how to improve vaccination rates, such making communication plans for kids with autism.

Even though it is directly referenced (p. 4, line 128), Figure 1 is not in the section about obstacles. It should be put in or taken out of the text. Table 1 in the policy section lists a few tools for assessing impairment, however it doesn't give any sources to back up the descriptions.
It would be helpful to include instances of how the applications are used. For instance, the WG-SS is used by South African national statistics agencies, while the MDS has been used in Chile to look at access to health care.
There are about 30 sources in the bibliography that are about the subject. It might be a good idea to add more references, especially some qualitative research that use interviews or focus groups.

Round 2

Reviewer 4 Report

Comments and Suggestions for Authors

I think my comments have been addressed appropriately. I don't have any further comments.

Author Response

Point-by-point response to Comments and Suggestions for Authors

Comments 1: I think my comments have been addressed appropriately. I don't have any further comments

Response 1: Thank you for your thorough review and for confirming that our revisions have adequately addressed your comments. We greatly appreciate the time and expertise you dedicated to improving our manuscript.

Reviewer 5 Report

Comments and Suggestions for Authors

These are the two recommended modifications: 1. In the section titled "Barriers and determinants of access." Subcategories of barriers should be rearranged in a more logical order, beginning with structural and physical barriers, followed by financial barriers, communication-related barriers, and lastly socio-cultural barriers. In an effort to prevent an excessively abstract listing, include a single concise, tangible study or example for each category. 2. In conclusion, Include a final brief paragraph that summarises the primary recommendations in bullet point format to facilitate reader comprehension and underscore the practicality of the proposed solutions.

Author Response

Point-by-point response to Comments and Suggestions for Authors

Comments 1: These are the two recommended modifications: 1. In the section titled "Barriers and determinants of access." Subcategories of barriers should be rearranged in a more logical order, beginning with structural and physical barriers, followed by financial barriers, communication-related barriers, and lastly socio-cultural barriers. In an effort to prevent an excessively abstract listing, include a single concise, tangible study or example for each category.

Response 1: Thank you for your helpful observation. We would like to highlight that the subsections in “Barriers and determinants of access” already adhere to the recommended sequence (structural and physical, financial, communication-related, and socio-cultural barriers). To further clarify this structure, we have added the following texts to the manuscript:

−       (lines 209-212) “For example, research in Kenya found that persons with physical disabilities faced significant challenges accessing COVID-19 vaccination sites due to inaccessible facilities, lack of ramps, narrow doorways, and inadequate accessible parking.”

−       (lines 221-222) “Research on people with visual impairment has specifically identified increased difficulties in accessing healthcare due to cost and lack of insurance coverage.”

−       (lines 226-232) “An exploratory qualitative study of medical specialists revealed a fundamental uncertainty about professional responsibility, with many questioning whether caring for people with intellectual disabilities was their job. The research found that stigma and limiting representations about the abilities of people with intellectual disabilities often lead professionals to generalize emergencies or legal exceptions to all patients in this population, systematically failing to obtain informed consent from the person themselves.”

−       (lines 236-238) “Research on cancer screening among deaf, deafblind, and hard-of-hearing adults demonstrated the critical need for specialized community health navigators to overcome communication barriers.”
(lines 251-254) “For example, research in Fiji documented how stigma and lack of disability-specific knowledge among healthcare providers created significant barriers to vaccination access for children with disabilities, contributing to vaccination rates of approximately half compared to national averages.”

These changes improve narrative flow and ground each barrier category in a real-world example, enhancing clarity. We trust this revision addresses your recommendation and strengthens the manuscript's practical focus.

Comments 2: 2. In conclusion, Include a final brief paragraph that summarises the primary recommendations in bullet point format to facilitate reader comprehension and underscore the practicality of the proposed solutions.

Response 2: Thank you for your insightful suggestion. We have retained the original conclusion paragraph in full and inserted the requested summary of primary recommendations immediately before the closing sentence (lines 379-387): “To translate our analysis into practice, we recommend:

·        Ensuring physical and organizational accessibility of all preventive health facilities.

·        Incorporating disability-specific training and accessible communication protocols for all health workers.

·        Actively involving PwD and their representative organizations in service design and policy planning.

·        Standardizing and sensitizing disability assessment tools to inform equitable resource allocation and monitoring.”

We trust this enhancement will further aid reader comprehension and underscore the applicability of our proposed solutions.